# ANOMALY DETECTION IN DYNAMICAL SYSTEMS FROM MEASURED TIME SERIES

## ABSTRACT

The paper addresses a problem of abnormalities detection in nonlinear processes represented by measured time series. Anomaly detection problem is usually formulated as finding outlier data points relative to some usual signals such as unexpected spikes, drops, or trend changes. In nonlinear dynamical systems, there are cases where a time series does not contain statistical outliers while the process corresponds to an abnormal configuration of the dynamical system. Since the polynomial neural architecture has a strong connection with the theory of differential equations, we use it for the feature extraction that describes the dynamical system itself. The paper considers both simulations and a practical example with real measurements. The applicability of the proposed approach and it's benchmarking with the existing methods is discussed.

## 1 INTRODUCTION

Most of the works related to anomaly detection in time series data are referred to the detection of the "observation which deviates so much from other observations as to arouse suspicions that it was generated by a different mechanism" (see Hawkins (1980)). Anomaly detection problem is usually formulated as finding outlier data points relative to some usual signals such as unexpected spikes, drops, trend changes, and level shifts. Blázquez-García et al. (2020) provides a literature review that deals exclusively with time series data and provides a taxonomy for the classification of outlier detection techniques according to their main characteristics.

In contrast to these methods, we address the problem of anomaly detection in dynamical systems from measured time series. In this case, we are interested in detecting the anomalies which do not deviate so much from other observations while they were still generated by a different configuration of the dynamical system. To better explain this issue, let us consider a dynamical system in the form of the ordinary differential equation (ODE):

$$\frac{d}{dt}\mathbf{X} = F(t, \mathbf{X}, a_1, a_2),\tag{1}$$

where $\mathbf{X} = (x_1, x_2, \ldots, x_n)$ is a state vector, $F$ is nonlinear function depending on two scalar parameters $a_1$ and $a_2$. The dynamical system (1) generates a trajectory in the form of the multivariate time series as a particular solution for a given initial condition $\mathbf{X}_0$. Let us also assume for simplicity that the initial conditions $\mathbf{X}(0) = \mathbf{X}_0$ are always the same for different trajectories but only parameters are varied and belong to the normal distribution.

Since the system (1) is nonlinear, the distribution of the trajectories is unknown in advance and may differ from the normal one. This may cause that trajectories corresponding to abnormal system parameters are located somewhere among the other trajectories. Fig. 1 demonstrates that parameters of the system (1) taken from the tail of the normal distribution can correspond to centered trajectories in the time-space.

This example formulates the problem of anomalies detection in the dynamical systems represented only by the measured trajectories. We are interested in the unsupervised methods for recovering the representative features of the dynamical system from the time series. Such a method should calculate features that are correlated with the dynamical system itself but not just with a time series that is generated by the dynamical system. Also, collecting massive training sets in industrial applications

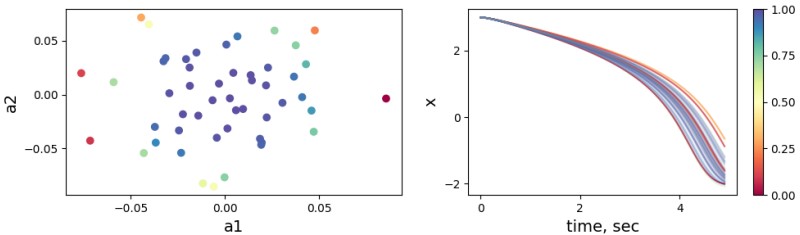

Figure 1: The abnormal parameters of the dynamical system from the tail of the normal distribution may correspond to centered trajectories in the time-space. The normal distribution is not preserved due to the nonlinear differential transformation that describes the dynamical system behavior.

is often not feasible. This makes it difficult to apply existing approaches that are based on either the extraction of the statistical features or state-of-the-art neural networks for temporary dependencies modeling. To address these lack-of-the-data challenges, we rely on the polynomial neural networks that were recently gain the attention of researchers in the field of learning of dynamical systems from measured data.

The most relevant to our research work is the paper of authors López et al. (1993), where the connection between the system of ODEs and polynomial neural networks (PNN) is introduced. Further, the PNN architectures were also widely highlighted in the literature. For example, Zjavka (2011) proposes a polynomial neural architecture that approximates differential equations. The Legendre polynomial is chosen as a basis by Yang et al. (2018). In all these papers, the authors suggest either integrating the parametrized polynomial ODEs or training PNN as a black-box model for identification of the dynamical systems from measured time series and prediction of new trajectories.

In opposite to these works, Ivanov et al. (2020) suggests a deterministic algorithm to translate an arbitrary nonlinear ODE to PNN without training. If the dynamics of the system follows approximately a given ODE, the Taylor mapping approach allows calculating weights of the neural network (TM-PNN) directly from the equations. This TM-PNN equals initial ODEs with the necessary level of accuracy and can be fine-tuned to recover true dynamics from real measurements without numerical integration of the ODEs. Further, Ivanov & Agapov (2020) applied TM-PNN architecture in a practical application to control one of the larges X-ray source. A deep TM-PNN is initialized from the ODEs that describe the charged particle motion and consists of more than 1500 hidden polynomial layers with unique weights that are fine-tuned with only one measured trajectory of the real system.

For the introduced problem of anomaly detection in the dynamical systems represented by time series, we utilize both PNN and TM-PNN architectures. Similar to López et al. (1993), we train PNN from scratch with measured trajectories. But instead of using the resulting model for dynamics prediction, we focus on the feature extraction problem and consider the weights of the PNN as features describing the dynamical system itself. Since this approach does not require any a priory knowledge about the dynamical system in the form of the ODEs and relies only on data, we compare it with statistical feature extraction and LSTM-based anomaly detection for time series.

For the benchmarking with a simple ODE-based parameters estimation, we utilize the TM-PNN architecture and follow the idea proposed in Ivanov & Agapov (2020). Given the system of ODEs that defines the dynamical system with an initial assumption of zero values for all free parameters, we translate the equation to PNN using the Taylor mapping technique. Then we compare the direct tuning of parameters of ODE with the fine-tuning of weights of the pre-initialized TM-PNN.

Though the Anonymous Company from the automotive industry provides the real measured data related to the introduced problem, the dataset is small and contains only 50 time series with only one true anomaly. Moreover, the anomaly scores of normal data are unknowns. For benchmarking purposes, we use simulated datasets of the same size based on the Van der Pol oscillator. Then we apply PNN for the analyses of real-world datasets that were generated in production settings.

## 2 SIMULATED DATA

In this section, we implement a simplified example with the Van der Pol oscillator. This equation is widely used in the physical sciences and engineering and can be used for the description of the pneumatic hammer, steam engine, periodic occurrence of epidemics, economic crises, depressions, and heartbeat. The equation has well-studied dynamics and is widely used for testing of numerical methods (see e.g. Pan & Duraisamy (2018)).

For simplicity, we parametrize the Van der Pol oscillator with two scalar parameters $a_1$ and $a_2$:

$$\begin{aligned} x' &= y, \\ y' &= y - (1 + a_1)x - (1 + a_2)x^2y, \end{aligned} \tag{2}$$

were $'$ means derivative on time, $(a_1, a_2) \in \mathcal{N}(0, \sigma I)$ with $\sigma = 0.001$ for clarity and I as identity two dimensional matrix.

For experiments with each method, we generate 100 datasets of 50 trajectories each for different sampled parameters $(a_1, a_2)$. Each trajectory is the numerical solution of the system (2) with constant initial condition $(x_0, y_0) = (3.0, 0)$ during 5 sec time interval with fixed time step 0.01 sec. To solve the system of ODEs we use the traditional Runge – Kutta method of forth order and then add white noise to all variables $x$ and $y$.

After this procedure, each trajectory is represented by two-variable time series $\{x_i, y_i\}$ with 500 stamps and corresponds to certain values of parameters $(a_1, a_2)$. The true score of abnormality for each trajectory is associated with the probability values of these parameters. Fig. 1 represents an example of the generated dataset. On the left plot, the result of the sampling the $(a_1, a_2)$ is demonstrated. The right plot shows $x$ values of each trajectory with a corresponding anomaly score. Since the system (2) is nonlinear, the normal distribution in parameter space is not preserved in time-space for trajectories. Namely, the true abnormal trajectories can be positioned in the arbitrary locations inside the bunch of all generated trajectories.

In the next section, we explore different methods for feature extraction from these time series data. Since extracted features have different physical meanings and units, we use the Isolation Forest algorithm to score trajectories in terms of abnormality. After scores are estimated and converted to ranks, we use both the Kendall rank correlation coefficient ($\tau$) and Spearman's rank correlation coefficient ($\rho$) to evaluate the method performance in comparison with true ranks associated with the probability values of parameters $(a_1, a_2)$. Both $\tau$ and $\rho$ metrics between two variables will be high when observations have a similar rank, and low when observations have a dissimilar rank.

We also introduce a heuristic accuracy metric. In each dataset of 50 samples, we consider a successful anomaly detection if the trajectory with the worst true score is detected within three less scored trajectories by the given method. As an accuracy, we calculate the fraction of such successful detections among all generated datasets.

The intuition of this heuristic metric is the following. Though the rank correlation coefficients are mathematically rigorous metrics, it is expected in the practical application that anomalies arise not very often. It may be more important to detect a true anomaly among trajectories with the lowest scores rather than rank normal trajectories correctly. Three trajectories with the lowest scores among 50 samples roughly correspond to the 5% threshold that is selected arbitrarily. In our opinion, the rank correlation coefficients along with such a heuristic accuracy demonstrate both the performance of the extracted features and the ability to detect anomalies in practical applications.

### 2.1 PNN ARCHITECTURE AND PNN-BASED FEATURES

Since each time series has 500 stamps and generated by a randomly defined dynamical system, we construct deep PNN architecture with 500 polynomial layers with shared weights. Each layer is described by the transformation

$$\mathcal{M} : \begin{pmatrix} x_{i+1} \\ y_{i+1} \end{pmatrix} = W_1 \begin{pmatrix} x_i \\ y_i \end{pmatrix} + W_2 \begin{pmatrix} x_i^2 \\ x_i y_i \\ y_i^2 \end{pmatrix} + W_3 \begin{pmatrix} x_i^3 \\ x_i^2 y_i \\ x_i y_i^2 \\ y_i^3 \end{pmatrix} \tag{3}$$

Figure 2: Deep PNN architecture with shared weights.

with initialized from uniform transform weights $W_1 = I, W_2 = 0, W_3 = 0$.

Fig. 2 represents a multioutput deep PNN architecture with shared weights that propagate initial values $(x_0, y_0)$ at the beginning of the trajectory along the time. Since it is indicated by López et al. (1993) that such architecture corresponds to a system of ODEs, we train a separate PNN for each trajectory and consider fitted weights $W_1, W_2, W_3$ as features for further anomaly scoring.

For the training, we use Adam optimizer with default parameters implemented in Keras/TensorFlow library. Though each layer $\mathcal{M}$ represents a local dynamics with a small time step, the multioutput architecture allows processing noisy trajectories considering loss function as mean squared error (MSE) for all time steps $(x_i, y_i)$ altogether.

## 2.2 STATISTICAL FEATURES (TSFRESH)

The benchmarking method for abnormal process detection in the industry is a threshold-based decision based on some statistical characteristics of the measured processes. The exact rules for anomaly detection can be adaptive and depend on the testing equipment specification but still depends on the statistical features.

To represent such a threshold-based anomaly detection, we use *tsfresh* library that was developed by Christ et al. (2016) and allows us to calculate a comprehensive number of features. After automatic feature extraction from the generated time series, we apply the classical principal component analysis (PCA) to preserve only 5 components. The range from 2 to 10 components was investigated but only a weak influence on the anomaly scores is explored.

Tabel 1 contains a comparison of scoring of trajectories based on statistical features extracted from time series and PNN-based weights that are fitted during 1000 epochs. Regardless of the accuracy and $(\tau, \rho)$ performance, statistical features sometimes disagree with true anomaly scores calculated from the distribution of parameters of the dynamical systems.

On the other hand, the minimum values of rank correlation coefficients are always positive for the PNN weights along with the higher accuracy value. This makes the PNN-based approach at least an alternative for industrial applications where threshold-based rules do not detect all anomalies in the production processes.

Table 1: Comparison of *tsfresh* and PNN

| METHOD | ACCURACY | METRIC | MEDIAN | AVG | STD | MIN | MAX |
|--------|----------|--------|--------|-----|-----|-----|-----|
| tsfresh | 0.59 | $\tau$ | 0.19 | 0.21 | 0.25 | -0.32 | 0.72 |
|         |      | $\rho$ | 0.28 | 0.26 | 0.30 | -0.40 | 0.85 |
| PNN | 0.80 | $\tau$ | 0.34 | 0.39 | 0.14 | 0.20 | 0.66 |
|     |      | $\rho$ | 0.44 | 0.47 | 0.16 | 0.23 | 0.79 |

## 2.3 TRADITIONAL NEURAL NETWORKS (LSTM AUTOENCODER)

Another possible solution for anomaly detection in time series is using recurrent or convolution neural networks. Unfortunately, most existing deep learning efforts in the time series anomaly detection are related to statistical outliers and do not tackle our problem formulation. For the addressed problem, only relatively small datasets of unlabeled data are available. Nonetheless, for benchmarking purposes, we apply the LSTM-based autoencoder to extract latent features of time series.

Since the LSTM neural network is a general-purpose architecture, it is not possible to train a separate neural network for each time series. Instead, one has to train LSTM on the whole training batch that causes an issue with an increasing number of training epochs in the considered task. For our simulated data, the LSTM-based autoencoder does not generalize data but memorize all trajectories. The increase in number of the latent variables from 5 to 15 did not improve the situation. The scoring performance is reduced while training epochs are increased (see Tabel 2).

Table 2: Memorization of the LSTM-AE with the increase of training epochs

| METHOD | ACCURACY | METRIC | MEDIAN | AVG | STD | MIN | MAX |
|---|---|---|---|---|---|---|---|
| LSTM-AE (1000 epochs) | 0.34 | $\tau$ | 0.14 | 0.17 | 0.37 | -0.46 | 0.70 |
| | | $\rho$ | 0.15 | 0.19 | 0.44 | -0.54 | 0.78 |
| LSTM-AE (5000 epochs) | 0.28 | $\tau$ | 0.08 | 0.16 | 0.20 | -0.15 | 0.52 |
| | | $\rho$ | 0.12 | 0.19 | 0.24 | -0.23 | 0.57 |

In opposite to this, PNN training demonstrates convergence with increased training epochs. One of the possible reasons for this is that the PNN architecture corresponds to some a priory unknown system of ODEs and during training with a single trajectory PNN more accurately recovers the physical properties of the dynamical system. A brief explanation of this property is presented in the discussion section. The theoretical examination along with the exploration of other general-purpose neural architectures are not covered by this research work.

Table 3: Convergence of the PNN with the increase of the training epochs

| METHOD | ACCURACY | METRIC | MEDIAN | AVG | STD | MIN | MAX |
|---|---|---|---|---|---|---|---|
| PNN (1000 epochs) | 0.80 | $\tau$ | 0.34 | 0.39 | 0.14 | 0.20 | 0.66 |
| | | $\rho$ | 0.44 | 0.47 | 0.16 | 0.23 | 0.79 |
| PNN (5000 epochs) | 0.92 | $\tau$ | 0.51 | 0.50 | 0.09 | 0.36 | 0.68 |
| | | $\rho$ | 0.60 | 0.60 | 0.11 | 0.40 | 0.82 |

## 2.4 EQUATION-BASED PARAMETERS FITTING

While in the sections 2.2 and 2.3 we train PNN from scratch, in this section, we take into account a priori knowledge about the dynamical system in the form of the ODEs. Also, we compare the fine-tuning of the pre-initialized PNN (TM-PNN) with the Taylor map and the direct fitting of ODEs' parameters.

The traditional fitting of the parameters $(a_1, a_2)$ in the equation (2) is implemented as a search procedure through numerical solving of the ODE with Runge–Kutta method of forth order. Given the generated trajectory, the parameters in the equation are identified as the solution of the inverse problem starting with the initial approach $(a_1, a_2) = (0, 0)$.

Since in the ODE-based fitting we parametrize equation and preserve its structure, let us also consider the equation (2) with $a_1 = 0$ and $a_2 = 0$ as a priori available knowledge about the dynamical system that were used to generate a given trajectory:

$$\frac{d}{dt}\mathbf{X} = \frac{d}{dt}\begin{pmatrix} x \\ y \end{pmatrix} = \begin{pmatrix} y \\ y - x - x^2 y, \end{pmatrix} = \begin{pmatrix} 0 & 1 \\ -1 & 1 \end{pmatrix}\begin{pmatrix} x \\ y \end{pmatrix} + \begin{pmatrix} 0 & 0 & 0 & 0 \\ 0 & -1 & 0 & 0 \end{pmatrix}\begin{pmatrix} x^3 \\ x^2 y \\ xy^2 \\ y^3 \end{pmatrix} = \tag{4}$$

$$P_1\mathbf{X} + P_3\mathbf{X}^{[3]},$$

where $\mathbf{X} = (x, y)$ and $\mathbf{X}^{[3]} = (x^3, x^2 y, xy^2, y^3)$. Following algorithm presented by Ivanov et al. (2020), one can calculate a Taylor map for this equation in the form (3)

$$\mathbf{X}_{i+1} = W_1\mathbf{X}_i + W_2\mathbf{X}_i^{[2]} + W_3\mathbf{X}_i^{[3]} \tag{5}$$

that in the combination with the equation 4 results in the ODE

$$\mathbf{X}'_{i+1} = P_1 W_1 \mathbf{X}_i + P_1 W_2 \mathbf{X}_i^{[2]} + \left( P_1 W_3 + P_3 W_1^{[3]} \right) \mathbf{X}_i^{[3]} + \mathcal{O}(\mathbf{X}_i^{[3]}), \tag{6}$$

where $\mathbf{X}^{[k]}$ means the $k$-th Kronecker power. Comparing (6) with (5) yields a new system of equations concerning the weight matrices:

$$\begin{aligned} W'_1 &= P_1 W_1, \\ W'_2 &= P_1 W_2, \\ W'_3 &= P_1 W_3 + P_3 W_1. \end{aligned} \tag{7}$$

The solution of the equation (7) with initial conditions $W_1(0) = I, W_2(0) = 0, W_3(0) = 0$ for the time interval $[0, T = 5]$ results a Taylor map $\mathcal{M}$ with matrices $W_1(T), W_2(T)$ and $W_3(T)$:

$$W_1 = \begin{pmatrix} 0.999959378500493 & 0.009040499726132 \\ -0.009040499726132 & 1.00899987822663 \end{pmatrix}, \quad W_2 = 0,$$

$$W_3 = \begin{pmatrix} 1.2204426e-07 & -4.0741619e-05 & -2.4518692e-07 & -5.5367007e-10 \\ 4.0741619e-05 & -0.0090803848 & -8.1971951e-05 & -2.4684793e-07 \end{pmatrix} \tag{8}$$

that describes the dynamics of the Van der Pol oscillator for $a_1 = 0$ and $a_2 = 0$. Since the calculated map $\mathcal{M}$ is valid for arbitrary initial conditions, the TM-PNN can be constructed by sequential concatenation of maps with shared weights.

The only difference between the TM-PNN and the PNN used in the previous section is the initial weights. As long as we compare the method with ODE-based fitting, we can incorporate initial knowledge in form of the system of ODEs inside the PNN by calculating the initial approach for weights (8). Further, these weights are fine-tuned with generated time series without solving ODE. For both approaches, we stop fitting when MSE for both variables in the trajectory becomes less than $10^{-5}$.

Table 4: Comparison of the ODE-based parameters fitting with the TM-PNN fine-tuning

| METHOD | ACCURACY | METRIC | MEDIAN | AVG | STD | MIN | MAX |
|--------|----------|--------|--------|-----|-----|-----|-----|
| ODE | 0.98 | $\tau$ | 0.50 | 0.52 | 0.05 | 0.48 | 0.59 |
| | | $\rho$ | 0.61 | 0.60 | 0.03 | 0.56 | 0.62 |
| TM-PNN | 0.96 | $\tau$ | 0.48 | 0.49 | 0.06 | 0.41 | 0.56 |
| | | $\rho$ | 0.55 | 0.57 | 0.04 | 0.53 | 0.63 |

## 3 EXPERIMENTAL DATA

The dataset consists of 50 two-dimensional time series with one true abnormal trajectory that is generated in the real-world production settings. Fig. 3 (left) demonstrates the time-space with the filtered trajectories for the first variable. The simple moving average is applied to remove noise from measurements. Note that the true scores of all normal trajectories are unknown and only information about the true abnormal trajectory is available.

To detect this anomaly within 50 trajectories, we apply the considered above methods and score the time-series based on the extracted features with the help of the Isolation Forest algorithm. Both *tsfresh*-based features and LSTM latent variables do not detect abnormal trajectory while the PNN-based features provide a meaningful result. The right plot in Fig. 3 presents scoring based on the PNN weights.

As a polynomial layer in the PNN, we use the second-order transformation (3) with the additional matrix $W_0$ for free terms in the Taylor map. Given the two-dimensional state vector of the measurements, the total amount of weights equals to 12. Fig. 4 shows the weights of each trajectory in parallel coordinates. The weights for the true abnormal curve differ from the rest of the trajectories and anomaly can be easily detected with the Isolation Forest approach within the space of the PNN weights.

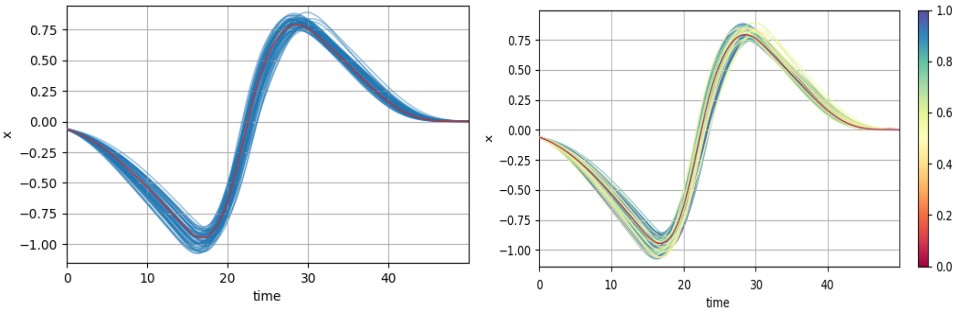

Figure 3: Input measured trajectories with a labeled true anomaly (left) and the same trajectories scored with PNN weights (right).

# 4 DISCUSSION

In the experiments, PNN-based features outperform both classical statistical features and latent variables from the LSTM autoencoder. Moreover, the performance of the TM-PNN is compatible with the simple benchmark of the fitting of the parameters of the ODE.

For the dynamical systems, the PNN is a suitable architecture because of its connection to differential equations. After fitting PNN with a single trajectory, PNN not only learns the given trajectory but approximates the dynamical systems itself. To explain this briefly, let us describe a simple way to go back from the fitted PNN to ODE.

Let us assume that after training PNN with a single trajectory, the weights equal to $W_1, W_2, W_3$ and define a polynomial layer in the form of the Taylor map (3). Let us also consider that this Taylor map corresponds to the discretization of the unknown ODE

$$\frac{t}{dt}\mathbf{X} = f(\mathbf{X})$$

with the Euler approximation

$$\mathbf{X}_{i+1} = \mathbf{X}_i + f(\mathbf{X})\Delta t. \tag{9}$$

Comparing (9) with (3) one can recover right-hand side of the differential equation using the fitted weights of the PNN:

$$\frac{t}{dt}\mathbf{X} = f(\mathbf{X}) = \left((W_1 - I)\mathbf{X} + W_2\mathbf{X}^{[2]} + W_3\mathbf{X}^{[3]}\right)/\Delta t.$$

Using different approximation schemes, one can obtain different exact formulas for the ODE. The main point is that the weights of the PNN correspond to the parameters of some system of the differential equations that is unknown a priori. This approach allows us to approximately reconstruct the system of ODEs that is supposed to generate the training trajectory without processing of ODEs with numerical solvers at all. Only weights of the PNN are tuned during training from time series.

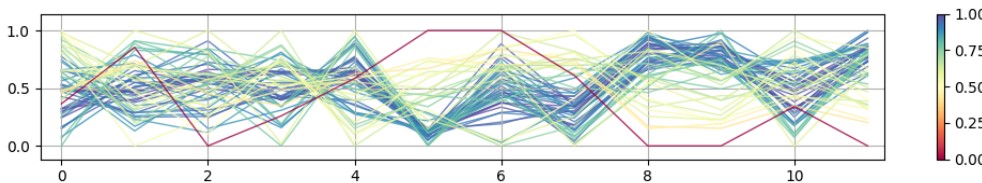

Figure 4: The PNN weights for each trajectory in parallel coordinates.

## 5 CONCLUSION

The PNN architecture for dynamical systems becomes a natural one in terms of the connection to systems of ODEs. It is demonstrated in the paper that weights of the PNN fitted from the time series can be considered as parameters of the initially unknown system of ODEs. Since ODE-based models are one of the most general approaches for describing the dynamical systems, the PNN-based representation learning outperform traditional methods for the introduced problem.

For the first set of experiments, we train PNN with zero initial weights from data without any initial knowledge about the dynamical system. PNN-based features outperform both classical statistical features and traditional neural networks. Also, it is still an open question on how to select a general-purpose neural architecture to solve the problem. On the other hand, the deep PNN provides a clear understanding of how to tune the number of hidden layers and orders of nonlinearities for weight matrices depending on the input time series.

For the second experiment, we incorporate initial knowledge about the system inside PNN using the Taylor mapping technique and compare the approach with the ODE-based parameters fitting. Both TM-PNN and ODE-based parametrization have a compatible performance. Though the ODE-based fitting slightly outperforms TM-PNN in the simulated experiment, it demonstrates that PNN can replace classical ODE-based parametrization when the system of ODEs is completely unknown or complex enough to be solved numerically in practice.

In the last example, we consider a real-world application of anomaly detection in production data. The use case is provided by Anonymous Company and represents a complex assembly testing in the automotive industry. Since nothing is known about the process in terms of the possible ODEs and the labeled anomaly is currently passed the existing testing equipment, the PNN can provide additional insights for anomaly detection and quality control in the production environment.

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
