# OpenReview forum: "Anomaly detection in dynamical systems from measured time series"
_ICLR.cc/2021/Conference — Reject_

### Official Review · AnonReviewer1 · 2020-10-23
**Relevant paper on anomaly detection, needs more details for clarity**

**Rating:** 4
**Confidence:** 4

**Review:**

The paper proposes the use of polynomial neural architectures (PNN) for anomaly detection in dynamical systems, and more precisely, for feature extraction. The choice of the architecture is motivated by the links between PNNs and ODEs that describe the evolution of dynamical systems.

Quality, clarity, originality and significance:
Pro: The paper is well written and approaches an important inverse problem, how to find anomalies in a dynamical system from some measured time series. The problem of anomaly detection is clearly significant, and the paper could benefit even more from a more extensive experimental setting on real world data. Cons: While the paper is mostly well argumented, I find that it leaves relevant information out that hinder its clarity as detailed below.

The main motivation of the paper is exemplified in Fig. 1 which shows that abnormal parameters of a dynamical system from the tail of the distribution might lead to normal trajectories, therefore misleading the anomaly detection. While this example is explaining already well the problem, I have a few questions. What are the ODEs generating this data, is this an example of the Van der Pol oscillator? If yes, it would be good to mention that. If it is a different system, is this system one-dimensional (x) or higher-dimensional? When PNN will be applied to this data, will it be applied on the fully observed system (let’s imagine two-dimensional with variables x and y) or will it be applied on the partially observed system (here x)? If the input to PNN will be only x then I believe the Figure is ok, because we try to find anomalies only from x and the plot on the right shows that normal trajectories still correspond to abnormal parameters. However, if the PNN will be applied on other variables in addition to x (let’s say x and y), then we would need to see the trajectories in this two-dimensional (or 3D if adding time) space. Because a trajectory can seem normal in one direction (here x) but appear as abnormal if we plot it in a different space. I believe the trajectories should be shown in the space of the data that will be used as input to PNN. In Sect. 2, is the full data (x and y) given to PNN always (this seems to be the case from eq 3 and Fig. 2)?

I believe the abstract would benefit from a sentence about the results.

Eq (1): Why only two parameters a1 and a2? If this hints to the Van der Pol oscillator, then X should be two-dimensional not higher-dimensional. Otherwise, if it is a general dynamical system, $\theta$ could be used for parameters. And then for illustration purposes as in Fig. 1 assume $\theta$/$a$ is two-dimensional.

What happens if the PNN architecture chosen in eq (3) changes? Are the results robust to the order of the polynomials?

Could the authors provide a reference for Isolation Forest and describe it a little bit and justify its choice?

Sect. 2.1: Do we need the same number of layers as the number of stamps? If yes, how to address scalability to real world applications?

Sect. 2.2: “calculate a comprehensive number of features” – how many features were computed and could the authors give some examples? Why is PCA needed on these features, and what is the impact of the number of PCs on the final results?

Would it be possible to combine the different tables of results into one table for an easier comparison? I am very much surprised by the difference in performance between the LSTM in Table 2 and PNN in Table 3. If LSTMs behave so poorly, are there other methods closer to PNN in performance?

Sect.2.4: Some aspects, eg eq (8) could be added to the appendix. Not sure I fully understand how the TM-PNN is constructed by “sequential concatenation”. Could the authors detail on this? What are the running times for ODE and PNN is Table 4?

Sect. 3: Would it be possible to have more details on the data? Some details are mentioned in the last paragraph from Conclusions, but could be moved here. What does this data represent, unless there are confidential issues? Eq (3) is a third order, not second order – which one is used here? Fig. 3, similar to Fig.1, should show the trajectories in the same space as the input to RNN, which might be more than one dimension (here x).

Fig. 4: I might be missing smth, but not sure I understand what are the parallel coordinates?

Related literature: there is a strong trend in ML in using Neural ODEs for modeling dynamical systems. Could the authors discuss the similarities and differences between Neural ODEs and PNNs?

---

### Official Review · AnonReviewer4 · 2020-10-28
**Official Blind Review #4**

**Rating:** 5
**Confidence:** 4

**Review:**

This paper is exploring anomaly detection problems where a time series does not contain statistical outliers. It shows the proposed polynomial neural network-based approach, which uses the fitted weights as extracted features, outperforms both classical statistical features and an LSTM autoencoder-based neural network.

It is indeed interesting and of practical value to distinguish any anomaly condition resulting from an abnormal configuration of the dynamical system, especially when it is difficult to accomplish using classical methods.

However, there are some places where this paper needs to further clarify:
- It is mentioned the proposed method will “train a separate PNN for each trajectory and consider fitted weights as features for further anomaly scoring”. If this is the case, I don’t see how two similar trajectories ($X,t$), with one deemed anomalous, would result in significantly different PNN weights based on equations (2), (3), and (9).
- The anomaly score needs to be clearly defined. From Figure 3, it seems a lower anomaly score indicates a higher chance of a trajectory being abnormal. However, I was thinking of the exact opposite while reading through the text.
- It should be further clarified that the color code on the left subplot of Figure 1 should be the probability of getting a specific ($a_1, a_2$) pair, rather than the anomaly score on the right subplot.
- Is it possible to use standard ODE-based fitting without any prior knowledge of the dynamic system (the $a_1=a_2=0$ assumption in Section 2.4)? If so, can this method be used in the last real-world application example of anomaly detection?

---

### Official Review · AnonReviewer3 · 2020-11-01
**Official Blind Review #3**

**Rating:** 4
**Confidence:** 5

**Review:**

This paper investigates the problem of anomaly detection in nonlinear processes. The main idea is to employ polynomial neural architecture for feature extraction that describes the dynamical system. The experiment results in both simulations and a practical example showed the effectiveness of the proposed technique.

Strengths:
+ Anomaly detection in dynamic systems is an interesting problem to study.
+ The proposed PNN architecture is technically sound.

Weaknesses:
- Several related works are not mentioned or compared.
- Only one practical dataset is used for evaluation.

My major concern is that many related works are not mentioned or compared in this work, e.g., One-class SVM[1], DSEBM[2], Deep SVDD[3], DAGMM[4], and Neural ODE [5].

[1] One-Class SVMs for Document Classification, JMLR, pp. 139-154, 2001

[2] Deep structured energy based models for anomaly detection. In International Conference on Machine Learning, pp. 1100–1109, 2016.

[3] Deep One-Class Classification, ICML 2018

[4] Deep Autoencoding Gaussian Mixture Model for Unsupervised Anomaly Detection, ICLR 2018

[5] Neural Ordinary Differential Equation, NIPS 2019

I would suggest the authors provide a comprehensive review of existing anomaly detection methods and discuss why they cannot address the anomaly detection problem in measured time series.

Another concern is that only one small dataset is used for evaluation. More real-world datasets should be used to justify that the proposed technique can be generalized to different use scenarios.

What's the computational complexity of PNN? What's the empirical running time?

PNN is only evaluated over two-dimensional time series, I am wondering whether it can be generalized to multivariate time series.

---

### Decision · Program_Chairs · 2021-01-07
**Final Decision**

**Decision:**

Reject

**Comment:**

The paper focuses on anomaly detection in dynamical systems from time series measurement. The originality of the contribution is to detect anomalies not based on the detection of OOD observations but from identified parameters or statistics of the dynamical system. They are using “polynomial neural networks. All the reviewers agree that the paper is not yet mature both in the form and in the technical content. The authors did not provide a rebuttal.